# Positive Psychology: Supervisor Leadership in Organizational Citizenship Behaviors in Nurses

**DOI:** 10.3390/healthcare10061043

**Published:** 2022-06-03

**Authors:** Nieves López-Ibort, Ana I. Gil-Lacruz, Luis Navarro-Elola, Ana C. Pastor-Tejedor, Jesús Pastor-Tejedor

**Affiliations:** 1Hospital Universitario Miguel Servet, 50009 Zaragoza, Spain; nlopezi@salud.aragon.es; 2Business Department, School of Engineering and Architecture, University of Zaragoza, 50018 Zaragoza, Spain; lnavarro@unizar.es (L.N.-E.); acpastor@unizar.es (A.C.P.-T.); jpastej@unizar.es (J.P.-T.)

**Keywords:** hospitals, workplace, leadership, organizational culture, power, health personnel

## Abstract

Introduction: In nursing, identifying factors encouraging positive work attitudes is extremely important since a nurse’s performance directly impacts the quality of the care they provide, and, therefore, their patients’ health. Objective: The main objective of this research is to analyze whether the supervisor–nurse relationship is positively correlated with a nurse’s organizational citizenship behaviors. Thus, we established a main hypothesis as follows: the quality of the supervisor–nurse interpersonal relationship is positively related to the job satisfaction of the nurse, controlled by moderating the effects of psychological empowerment, the perceived organizational support, and leader–leader exchange. Methodology: This is a cross-sectional descriptive study with individuals as the units of analysis. The population studied comprised all the nurses and supervisors working in nine public hospitals in the autonomous community of Aragon (Spain). The sample consisted of 2541 nurses, 192 supervisors, and 2500 paired dyads. Self-report questionnaires were used to ensure workers’ anonymity. The dependent variable was the nurse’s organizational citizenship behaviors; the main independent variable was the supervisor’s leadership; the moderating variables were the nurse’s empowerment, the organizational support the nurse perceived, and the quality of the supervisor–superior relationship. Results: Empirical evidence demonstrates that the quality of the supervisor–nurse relationship is positively correlated with organizational citizenship behaviors. The results also confirm the moderating effect of nurses’ empowerment and of the organizational support they perceive. Discussion: Our research shows how important it is for organizations to establish management practices promoting high-quality nurse–supervisor relationships; thus, hospital management should monitor both the supervisors’ performance and leadership. Conclusions: The quality of the relationship the supervisor establishes with their nurses is vitally important since it is a necessary requirement for beneficial results for the organization as a result of citizenship behavior practice.

## 1. Introduction

In the context of public hospitals in the Spanish autonomous community of Aragon, nursing supervisors play a challenging yet essential role to ensure nurses participate actively at work. The importance of this position lies in the strong, visible leadership a supervisor must exercise, which is based on two aspects: firstly, creating healthy environments in which to provide quality care; secondly, the positive influence they must exert on the attitudes of the nurses they manage.

Nurses’ work attitudes are characterized by organizational citizenship behaviors that improve organizational performance without seeking compensation. In fact, organizational citizenship behavior (OCB) [1] goes beyond the employee’s specified role. Moreover, as it is the person’s discretional behavior, it is less likely to be recognized by job descriptions and formal reward systems. OCB can be broken down into behaviors toward individuals, for example, altruism and courtesy, and behaviors toward the organization, for example, civic virtue, loyalty, and sportsmanship [2]. Empathic concern may form part of the context in which affective responses react to a relational conflict in the guise of OCB [3].

The main objective of this research is to study how nursing supervisors may reinforce nurses’ organizational citizenship behaviors. To define a model that takes into account simultaneously the complexity of this causal link, in this article we take into account how nurses value their empowerment, the organizational support and the relationship among leaders. On one hand, studies on structural empowerment in health centers show that positive changes in the structure can lead to healthier employees and improved organizational results, including better patient care. Psychological empowerment is especially important in health settings, where there have been cutbacks and morale is low [4]. On the other hand, management practices in healthcare, which provide substantial support for the professional practice of nursing, result in an increase in organizational citizenship behaviors [5]. In addition, the supervisor’s ability to influence the work of their lower-level workers is impacted by the quality of the relationship they have with their direct superior [6].

Examining this relationship in a health context is important as these behaviors can prove vital for patient satisfaction [7,8]. In this regard, there are four main contributions of this work to the state of art. First, the leader–member exchange is usually only measured from the viewpoint of subordinates. In this study, this relationship has been considered from the viewpoint of both the supervisor and the subordinate nurse [9]. Second, the literature review focuses especially on the performance of individual work and ignores the traditional concepts of work performance quantity or quality, such as: organizational citizenship behavior, pro-social organizational behavior, organizational spontaneity, contextual performance, or extra behavior, in other words, going above and beyond their duties [10]. Third, although the impact of the leader–member exchange on attitudes toward the organization and work has been demonstrated, not as much attention has been paid to variables moderating the results [9,11]. Fourth, few studies on nursing have addressed human relationships within health organizations [12,13,14,15] Furthermore, most of this empirical evidence only covers the English-speaking world [16].

## 2. Objective

The main objective of this research is to analyze whether the supervisor–nurse relationship is positively correlated with a nurse’s organizational citizenship behaviors. Thus, we established a main hypothesis as follows: The quality of the supervisor–nurse interpersonal relationship is positively related to the job satisfaction of the nurse, controlled by the moderating effects of psychological empowerment, the perceived organizational support and leader–leader exchange (Hypothesis 1).

**Hypothesis** **1(A).***The empowerment of the nurse plays a positive moderator role on the relationship between leader–member exchange and the organizational citizenship behaviour*.

**Hypothesis** **1(B).***The organizational support perceived by the nurse plays a positive moderator role on the relationship between leader–member exchange and the organizational citizenship behaviour*.

**Hypothesis** **1(C).***The leader–leader exchange plays a positive moderator role on the relationship between leader–member exchange and the organizational citizenship behaviour*.

Therefore, the link of the supervisor–nurse relationship with the nurse’s organizational citizenship behaviors has been analyzed to see whether it is moderated by the nurses’ empowerment, perceived organizational support, and leader–leader exchanges (See Figure 1).

## 3. Methodology

To that end, we carried out a cross-sectional descriptive study with individuals as the units of analysis. The population studied comprised all the nurses and supervisors working in nine public hospitals in the autonomous community of Aragon (Spain). Inclusion criterion was defined as a nurse/supervisor relationship of at least one month. The research team leader selected one or two nurses of each hospital who were responsible for collecting the questionnaires of their organizations. Participants were provided with a hard copy and an envelope that could be sealed without contact information, so their anonymity was guaranteed.

The hospitals are categorized by size into small and large (small ≤500 beds and large ≥501 beds), following the criterion of the Spanish Ministry for Health, Social Services, and Equality [17].

The unit researched in this study was defined as the nurse–supervisor dyad with a relationship lasting at least one month in general public hospitals in Aragon [18].

The sample consisted of 2541 nurses, 192 supervisors, and 2500 paired dyads. Self-report questionnaires were used to ensure workers’ anonymity (See Figure 2).

The following sociodemographic variables were included in our study as control variables: nurse age and gender; length of service as a nurse; length of service in that hospital; length of service working in that unit; length of the relationship with that supervisor; academic qualification; type of working day; number of nurses reporting directly to the supervisor; hospital size; size of the population where the hospital is located.

The quality of the nurse–supervisor relationship, measured with LMX-7, was taken as the independent variable. LMX is framed within a dyadic relationship by definition, but most studies proceed only with the viewpoint of the members in the LMX [9,19]. This research takes both perceptions into account: the nurse’s and the supervisor’s. The questionnaire by Graen and Uhl-Bien [20] was followed. It comprises seven items and a Likert scale with five response items (1 = Rarely; 5 = Very often).

The nurse’s organizational citizenship behaviors were considered as a dependent variable, measured by the questionnaire by Podsakoff et al. [21], which has twenty-four items, five for four of the dimensions (honesty, sportsmanship, courtesy, and altruism) and four for civic virtue. The items are rated on a Likert scale with seven response options (1 = Strongly disagree; 7 = Strongly agree).

The moderating variables studied in this research were as follows.

Empowerment (of the nurse), measured with Spreitzer’s questionnaire [22] (1995), which comprises thirteen items. The items are rated on a Likert scale with seven response options (1 = Not much; 5 = A lot).

Perceived organizational support (of the nurse), measured with the questionnaire by Eisenberger et al. [23]. The seventeen items are rated on a Likert scale with seven response options (1 = Strongly disagree; 7 = Strongly agree).

Quality of the supervisor–supervisor’s direct superior relationship (LLX), which is measured using the same criteria as LMX (See Table 1).

With the database ready, descriptive statistics of the variables and regression equations were carried out. The significance of the variables to the explanation of the organizational citizenship behavior of the nurse was analyzed using a multiple linear regression model with a step-by-step method, which is able to identify the effect of each variable while avoiding the problem of multicollinearity. At each step, the significance of the equation is studied to avoid the introduction of variables related to those already in the equation (collinearity), this results in a model that represents the best possible regression equation. After that, we investigated the statistical significance of the selected moderator variables (95% level) through a new regression equation. Each regression shows the interaction between each one-moderator variable (EMP, POS, LLX) and the independent variable LM(x).

## 4. Results

Below are the descriptive statistics with the means transformed into a scale ranging from 0 to 10 to show the differences between the scales. The variables with the higher scores are OCBs, nurse empowerment, leader–member exchange, leader–leader exchange, and, lastly, POS. For both large and small hospitals, OCB and POS are again the variables rated the best and worst, respectively, while there are slight differences between variables with an intermediate score (See Table 2).

H1: The correlation between the supervisor–nurse relationship (LMX(m)) and organizational citizenship behaviors (OCBs) is positive and significant, showing that the higher the nurse perceives the quality of the relationship (high scores in LMX(m)), the higher their organizational behaviors are. This first research hypothesis is confirmed (r = 0.142, *p* < 0.01).

This result is confirmed by the value obtained for the following equation
OCB = 5.20 + 0.11 · LMX(m) + *e*(1)
with an explained variance (R^2^) percentage of 3.4%.

The multiple linear regression model is used to study the moderation hypotheses. For that purpose, the following are entered into the equation: the dependent variable (OCB); the independent variable (LMX(m)); the moderating variable (empowerment, POS, or LLX, depending on the case); the product of the independent variable multiplied by the moderating variable. The significance of the latter term will indicate whether the studied variable is actually a moderating variable.OCB = CONSTANT + A1 **·** LMX(m) + A2 **·** MODERATING VARIABLE + A3 **·** (LMX(m) **·** MODERATING VARIABLE) + *e*

This structure is repeated three times independently based on the moderating variable:

H1(A). Empowerment: A3 = 0.08

With a variance percentage of 10.7% and a significant product coefficient (t = 4.56 and *p* < 0.001), the moderating effect of empowerment is confirmed in the relationship between LMX (m) and organizational citizenship behaviors; therefore, hypothesis H1(A) is supported.

H1(B). Perceived Organizational Support (POS): A3 = 0.06

With a variance percentage of 5.3% and a significant product coefficient (t = 5.58 and *p* < 0.001), the moderating effect of perceived organizational support is confirmed in the relationship between LMX (m) and organizational citizenship behaviors, thereby confirming hypothesis H1(B).

H1(C). Leader–Leader Exchange (LLX): A3 = 0.03

The percentage of variance rises to 3.6% and the coefficient of the variable product is not significant (t = 0.74 and *p* > 0.05). In this case, the moderating effect of LLX cannot be confirmed in the relationship between LMX (m) and organizational citizenship behaviors; therefore, hypothesis H1(C) is rejected.

Lastly, the results are analyzed based on hospital size, except for hypothesis H1(C), which has already been rejected. The overall significance of the exchange is more powerful for large hospitals. The significant moderation of perceived organizational support, which is significant overall, is also stronger for large hospitals. The same occurs with empowerment; the moderation is significant, but with an even greater difference for large hospitals (See Table 3).

We conclude this section with Table 4, which summarizes the main results. (See Table 4).

## 5. Discussion

As other studies have with samples of nurses, our research confirmed the positive relationship between the quality of the LMX and OCBs [7]. Concerning the same relationship, Chen et al. [8] found that the LMX of nursing personnel had a positive impact on nurses’ OCBs, scored by the supervisor, through the mediating variables of perceived trust and supervisor support.

Concerning hypothesis H1(A), research relating LMX and OCB with empowerment is limited; however, authors are unanimous in identifying a significant and positive relationship between empowerment and nurses’ organizational citizenship behaviors [24,25].

Our findings coincide with the results of Harris, Wheeler, and Kacmar [26], both in empowerment’s moderating role in the relationship between LMX and organizational citizenship behaviors, and in the way it moderates. As in their study, the highest OCB levels in our sample were obtained when LMX quality was high and empowerment was low.

When itemizing the hospitals by size, the moderating effect is greater in the large hospitals. These data suggest that empowerment can be more effective at increasing organizational citizenship behaviors in nurses in large hospitals.

Concerning hypothesis H1(B), referring to the moderating effect of POS in the relationship between the LMX and OCBs, as in the studies by Ehigie and Otukota [27] and Young [28], we confirm in our research that workers’ perception of the organization supporting them and being concerned about them is positively related to organizational citizenship behaviors. These feelings of obligation were also corroborated in previous studies with samples of nurses [29]. There are no interesting results when itemizing by hospital size.

To end this discussion, we did not find any evidence of a significant relationship between the leader–member exchanges and the leader–leader exchanges for hypothesis H1(C). In principle, it would seem reasonable to assume that when a supervisor has a high-quality relationship with their direct superior and, therefore, receives resources to distribute, the relationship with their nurses would improve.

This unexpected finding suggests that it may not be possible to extrapolate the results found by Sluss, Klimchak, and Holmes [6] to these organizations. In their research, the negative effects of less quality in the leader–member exchange were mitigated by greater quality in the leader–leader exchange.

Recent empirical evidence confirms the positive relationship between employees’ behaviors and their performance [30]. In the specific field of nursing, identifying factors encouraging positive work attitudes is extremely important since a nurse’s performance directly impacts the quality of their care and, therefore, their patients’ health and wellbeing [31].

Our research shows how important it is for organizations to establish management practices promoting high-quality nurse–supervisor relationships and that hospital management should not only monitor the performance of their supervisors, but also the leadership they provide, and the relationships they maintain with their nurses. In fact, policy leadership should go hand in hand with guiding professionals with appropriate training in the appropriate techniques to achieve the required outcomes [32].

Given the importance of these relationships, supervisors should be evaluated in terms of their relationships with their subordinates when examining their leadership-related behavior. Consequently, since the quality of the supervisor–nurse relationship has become another indicator of a supervisor’s performance, it should be given the same weight as other current indicators, and management’s actions should be based on the results.

Supervisors that do not mirror management’s leadership should be identified so they can be trained in these skills or dismissed, otherwise, they will contaminate one or several of the organization’s units, thereby impacting on the nurses’ responses and, as a result, users’ health.

Similarly, strong and positive exchanges between nurses and their hospitals urgently need to be fostered. Nurses need to see their hospitals making discretional gestures so that these professionals feel looked after, recognized, and supported.

One of this study’s limitations is its cross-sectional design since leader–member exchanges are dynamic, develop over time, and, therefore, can change. A static approach to the data makes it impossible to offer sound inferences of the direction of causality, which could be gleaned from the evidence of covariation in the study variables and theoretical associations.

## 6. Conclusions

The following conclusions can be reached using the results of this research as a reference.

The quality of the relationship the supervisor establishes with their nurses is vitally important since it is a necessary requirement for beneficial results for the organization as a result of citizenship behavior practice.

The moderating variables considered were empowerment, perceived organizational support, and the relationship between supervisors and their direct superiors. The first two are valuable variables, while there is no conclusive empirical evidence for the third.

A larger hospital size (measured by the number of beds) is an important variable compared with organizational citizenship behaviors and with the three moderating variables, which do not negatively penalize the accepted hypotheses.

## Figures and Tables

**Figure 1 healthcare-10-01043-f001:**
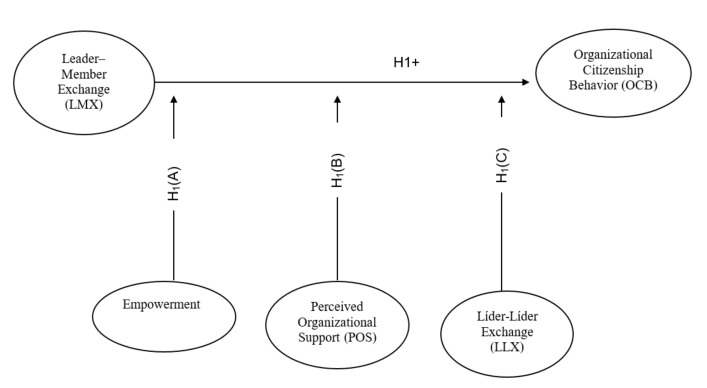
Theoretical model. Figure made by the authors.

**Figure 2 healthcare-10-01043-f002:**
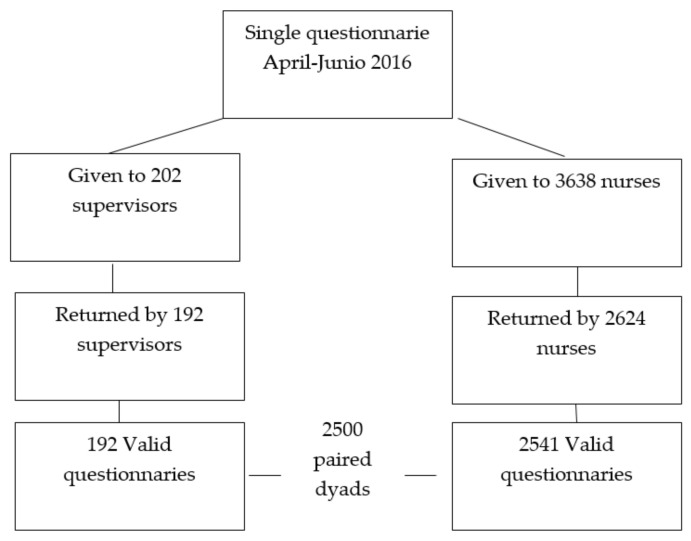
Study flow chart.

**Table 1 healthcare-10-01043-t001:** Summary of scales used in this research.

Variable	Scale	Author	Cronbach’s Alpha
Empowerment	Psychological Empowerment Instrument	[22] Spreitzer (1995)	0.881
Organizational Citizenship Behavior	Scale of Organizational Citizenship Behavior	[10] Podsakoff, Mackenzie, Moorman, and Fetter (1990)	0.773
Perceived Organizational Support	Survey of Perceived Organizational Support	[23] Eisenberger, Huntington, Hutchison, and Sowa (1986)	0.938
Leader–Member Exchange (nurse’s viewpoint)	LMX-7	[20] Graen and Uhl-Bien (1995)	0.925
Leader–Member Exchange (supervisor’s viewpoint)	LMX-7	[20] Graen and Uhl-Bien (1995)	0.920
Leader–Leader Exchange	LMX-7	[20] Graen and Uhl-Bien (1995)	0.947

**Table 2 healthcare-10-01043-t002:** Description of the questionnaires’ descriptive statistics by hospital size (range 0–10).

Scale	Hospital Size				Total		*p*-Value
	Small		Large				
	Mean	S.D.	Mean	S.D.	Mean	S.D.	
Empowerment	6.86	1.34	6.63	1.40	6.71	1.38	0.03 *
OCB	7.64	0.79	7.63	0.80	7.64	0.80	0.00 *
POS	4.29	1.61	3.65	1.64	3.88	1.66	0.12
LMX (m)	6.61	1.92	6.22	2.14	6.36	2.07	0.07
LMX (l)	6.84	1.77	6.64	1.78	6.71	1.78	0.01 *
LLX	7.43	1.84	6.01	2.18	6.53	2.17	0.23

S.D. = standard deviation; * *p*-value < 0.05.

**Table 3 healthcare-10-01043-t003:** Coefficients and significance of moderations for the LMX relationship and overall OCB depending on hospital size.

		Initial		EMP		POS		LLX	
		Coef.	*p*-Value	Coef.	*p*-Value	Coef.	*p*-Value	Coef.	*p*-Value
Overall		0.107	0.03 *	0.083	0.01 *	0.059	0.01 *	0.003	0.08
Size	Large	0.015	0.00 *	0.147	0.04 *	0.081	0.03 *	-	
	Small	–0.005	0.07	0.090	0.00 *	0.045	0.00 *	-	

* *p*-value < 0.05.

**Table 4 healthcare-10-01043-t004:** Summary of the main results.

Hypothesis	Overall	Hospital Size	
		Large	Small
H1	Confirmed	Confirmed	Rejected
H1(A)	Confirmed	Confirmed	Confirmed
H1(B)	Confirmed	Confirmed	Confirmed
H1(C)	Rejected	-	-

## Data Availability

Due to the nature of this research, participants of this study did not agree for their data to be shared publicly, so supporting data are not available.

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
