# Peer review of "Positive Psychology: Supervisor Leadership in Organizational Citizenship Behaviors in Nurses"

_healthcare, 2022, doi:10.3390/healthcare10061043_

Round 1
Reviewer 1 Report
It's ok
Reviewer 2 Report
Dear Authors!
Thank you for exploring a very interesting and important topic of Positive Psychology: Supervisor Leadership in Organizational Citizenship Behaviors in Nurses.
I am glad you have made an attempt to improve the manuscript, and my previous remarks have been incorporated; in its current form, it presents a much better quality compared to the previous versions. I can recommend it for publication.
Best wishes,
The reviewer.
This manuscript is a resubmission of an earlier submission. The following is a list of the peer review reports and author responses from that submission.
Round 1
Reviewer 1 Report
Match keywords to MeSH terms.
The sections or sections of the abstract must be the same as in the main body of the study.
In line 31, refer to where you got that data from.
You should develop the introduction much more by talking about the subject of your study.
Lines 36-38 are not part of the introduction.
Figure 1 (and all) refer to where you got it from or if it is your own.
2.- Moderating variables. should be part of the methodology.
Lines 61-72 should not be part of the introduction. Include it in the section "weaknesses and strengths" in section: results.
After defining the main objective, the hypotheses that it poses later should appear.
Explain what inclusion criteria you used for your sample size; how he defined himself (n=) and how he recruited the participants (random, for convenience, etc).
From line 106, the RESULTS section should appear where you describe the results (without personal opinion) of your study.
Resize tables to match text wrapping.
In methodology, you must define which statistics you are going to use (t= t Studet???,) and above all define their level of significance (pValue) before they appear in the results.
In the last column of Table 2, the P values ​​should appear regardless of whether or not they are statistically significant.
You must include the doi or access url to all the bibliographical references you consulted
Author Response
Thank you very much for your kind words, and for your effort in reviewing the paper. We understand your point of view and we value your comments as very accurate and valuable. They have helped us to improve our paper and we hope you find the changes suitable.
- Match keywords to MeSH terms.
Previous keywords have been replaced by the following ones: hospitals; workplace; leadership; organizational culture; power; health personnel.
- The sections or sections of the abstract must be the same as in the main body of the study.
The abstract has been restructured with the same sections as in the main text: Introduction, Objective, Methodology, Results, Discussion and Conclusions.
- In line 31, refer to where you got that data from.
We have specified that we have obtained the data from the public hospitals of the Spanish Autonomous Community of Aragon.
- You should develop the introduction much more by talking about the subject of your study. Lines 36-38 are not part of the introduction.
We agree that in the way we presented the information it was confused; we have rewritten the paragraphs to make the introduction clear.
- Figure 1 (and all) refer to where you got it from or if it is your own.
We have stated that figure 1 is our own.
- Moderating variables. should be part of the methodology.
Moderating variables have been explained in the methodology.
- Lines 61-72 should not be part of the introduction. Include it in the section "weaknesses and strengths" in section: results.
We have rewritten them in order to make the subject of the paper clear.
- After defining the main objective, the hypotheses that it poses later should appear.
Done.
- Explain what inclusion criteria you used for your sample size; how he defined himself (n=) and how he recruited the participants (random, for convenience, etc).
As indicated in the article, the population studied comprised all the nurses and supervisors working in nine public hospitals in the autonomous community of Aragon (Spain). In addition, we provided the following information: As inclusion criteria was defined as a nurse/supervisor relationship of at least one month. The research team leader selected one or two nurses of each hospital who were responsible for collecting the questionnaires of their organizations. Participants were provided with a hard copy and an envelope that could be sealed without contact information, so their anonymity was guaranteed.
- From line 106, the RESULTS section should appear where you describe the results (without personal opinion) of your study.
It has been verified that the results are free from the opinions of the authors and they are limited to the description of the results.
- Resize tables to match text wrapping.
Done.
- In methodology, you must define which statistics you are going to use (t= t Studet???,) and above all define their level of significance (p-Value) before they appear in the results.
With the data base ready, descriptive statistics of the variables and regression equations were carried out. The significance of the variables to the explanation of the organizational citizenship behaviour of the nurse was analysed using a multiple linear regression model with a step-by-step method which is able to identify the effect of each variable whilst avoiding the problem of multicollinearity. At each step, the significance of the equation is studied to avoid the introduction of variables related to those already in the equation (collinearity), this results in a model that represents the best possible regression equation. After that, we research about the statistical signification of the selected moderator variables (95% level) through a new regression equation. Each regression shows the interaction among each one-moderator variable () and the independent variable LM(x).
- In the last column of Table 2, the P values should appear regardless of whether or not they are statistically significant.
Done.
- You must include the doi or access url to all the bibliographical references you consulted
Done.
Reviewer 2 Report
Dear Authors!
Thank you for taking up a very interesting and important topic of Positive Psychology: Supervisor Leadership in Organizational Citizenship Behaviors in Nurses. Unfortunately, due to the fact that the weaknesses outweigh the strengths of the paper, I cannot recommend it for publication.
Improvement requires first of all, the theoretical part, as well as the research methodology description and the presentation of the results.
Below is a list of the main issues that need improvement:
1) The format of the references is not compatible with the journal requirements.
2) The introduction is poor. The theoretical positioning should be developed (so far there are only 5 references for this part). It definitely needs to be rewritten.
3) When presenting the figure of the theoretical model used (page 2), it is not clear whether it is the one proposed by the authors or if it comes from the literature. In the latter case, a citation would be appropriate.
4) More details are needed on sample selection. Was it random or maybe a non-random selection? Was it a purposive selection or maybe the snowball method as used? It is not clear right now.
5) There are no precise references in Table 1. In the "author" column, apart from the author's name, the year of publication should also be indicated.
6) The section „results” is missing. The „discussion” section appears immediately after "methodology".
7) The results of the study are presented in a rather chaotic way. Hypotheses should be derived as a result of a literature review, and then accepted or rejected and discussed. Currently, the first mention of the hypothesis appears on page 9, together with the information about its confirmation. It is confusing to the reader.
8) Are there any implications (both theoretical and practical) and maybe any limitations? It is worth listing them.
I encourage you to correct the paper and re-submit it.
Best wishes,
The reviewer.
Author Response
Thank you very much for your kind words, and for your effort in reviewing the paper. We understand your point of view and we value your comments as very accurate and valuable. They have helped us to improve our paper and we hope you find the changes suitable.
- The format of the references is not compatible with the journal requirements.
The format of the references has been adapted to the journal requirements. We have included the doi or url that give access to the cited articles.
- The introduction is poor. The theoretical positioning should be developed (so far there are only 5 references for this part). It definitely needs to be rewritten.
The introduction has been rewritten and we confirm that a total number of 16 references has been included.
- When presenting the figure of the theoretical model used (page 2), it is not clear whether it is the one proposed by the authors or if it comes from the literature. In the latter case, a citation would be appropriate.
We have stated that figure 1 is our own.
- More details are needed on sample selection. Was it random or maybe a non-random selection? Was it a purposive selection or maybe the snowball method as used? It is not clear right now.
As indicated in the article, the population studied comprised all the nurses and supervisors working in nine public hospitals in the autonomous community of Aragon (Spain). In addition, we provided the following information: As inclusion criteria was defined as a nurse/supervisor relationship of at least one month. The research team leader selected one or two nurses of each hospital who were responsible for collecting the questionnaires of their organizations. Participants were provided with a hard copy and an envelope that could be sealed without contact information, so their anonymity was guaranteed.
- There are no precise references in Table 1. In the "author" column, apart from the author's name, the year of publication should also be indicated.
Done
- The section „results” is missing. The „discussion” section appears immediately after "methodology".
We have split the section “Methodology” into two parts, that the second part has been entitled “Results”.
- The results of the study are presented in a rather chaotic way. Hypotheses should be derived as a result of a literature review, and then accepted or rejected and discussed. Currently, the first mention of the hypothesis appears on page 9, together with the information about its confirmation. It is confusing to the reader.
We have introduced major changes on the main text. It has been restructured with the following sections: Introduction, Objective, Methodology, Results, Discussion and Conclusions. The section “Introduction” presents the literature review, the section “Objective” the main objective and corresponding hypothesis, the section “Results” just focuses on main estimated coefficients and lastly in the section “Discussion” we compare our results to those obtained in previous research.
- Are there any implications (both theoretical and practical) and maybe any limitations? It is worth listing them.
In relation to the implications, we had included the following paragraphs:
Our research shows how important it is for organizations to establish management practices promoting high-quality nurse–supervisor relationships and that hospital man-agement should not only monitor the performance of their supervisors, but also the lead-ership they provide, and the relationships they maintain with their nurses. In fact, policy leadership should go hand in hand with guiding professionals with appropriate training in the appropriate techniques to achieve the required outcomes.32
Given the importance of these relationships, supervisors should be evaluated in terms of their relationships with their subordinates when examining their leader-ship-related behavior. Consequently, since the quality of the supervisor–nurse relation-ship has become another indicator of a supervisor’s performance, it should be given the same weight as other current indicators, and management’s actions should be based on the results.
Supervisors that do not mirror management’s leadership should be identified so they can be trained in these skills or dismissed. Otherwise, they will contaminate one or several of the organization’s units, thereby impacting on the nurses’ responses and, as a result, users’ health.
Similarly, strong and positive exchanges between nurses and their hospitals urgently need to be fostered. Nurses need to see their hospitals making discretional gestures so that these professionals feel looked after, recognized, and supported.
Regarding the limitations, we pointed out the following one:
One of this study’s limitations is its cross-sectional design, since leader–member ex-changes are dynamic, develop over time and, therefore, can change. A static approach to the data makes it impossible to offer sound inferences of the direction of causality, which could be gleaned from the evidence of covariation in the study variables and theoretical associations.
Round 2
Reviewer 1 Report
After reviewing the suggested modifications the article is potentially publishable. Receive cordial greetings.
Author Response
Thank you very much for the review effort.Reviewer 2 Report
Dear authors,
Thank you for considering my comments and providing a revised version of your paper. In my opinion, its scientific value increased significantly compared to previous version.
However, the manuscript still needs some editing fixes. For example, in the abstract (line 24) something like this appears: "Discussion: XXXXX". Review the text carefully to eliminate typos and other errors.
The most important issue is that the objective of the research in the ‘Objective’ section /lines 97-100/ differs from that of the abstract / lines 12-13/ Therefore, in 'Objective' section, I propose to describe it as follows.
“The main objective of this research is to analyze whether the supervisor–nurse relationship is positively correlated with a nurse’s organizational citizenship behaviors.
Thus, we established main hypothesis as follows: The quality of the supervisor-nurse interpersonal relationship is positively related to the job satisfaction of the nurse controlling by moderating effects of psychological empowerment, perceived organisational support and Leader-Leader Exchange (Hypothesis 1).”
I think it would be much clearer for the reader.
Hypotheses 1A, 1B, and 1C presented in the "Objective" section (lines 102-111) are identical. This needs to be corrected.
Regards,
The reviewer
Author Response
Thank you very much for the review. We understand your comments, and we have proceeded to correct them paying special attention to typos and other errors.
For example, in the abstract (line 24) something like this appears: "Discussion: XXXXX". Review the text carefully to eliminate typos and other errors.
We have completed the section devoted to the Discussion in the abstract as it follows:
Discussion: Identifying factors encouraging positive work attitudes is extremely important since a nurse’s performance directly impacts their patients’ wellbeing.
The most important issue is that the objective of the research in the ‘Objective’ section /lines 97-100/ differs from that of the abstract / lines 12-13/ Therefore, in 'Objective' section, I propose to describe it as follows. “The main objective of this research is to analyze whether the supervisor–nurse relationship is positively correlated with a nurse’s organizational citizenship behaviors. Thus, we established main hypothesis as follows: The quality of the supervisor-nurse interpersonal relationship is positively related to the job satisfaction of the nurse controlling by moderating effects of psychological empowerment, perceived organisational support and Leader-Leader Exchange (Hypothesis 1).” I think it would be much clearer for the reader.
We have proceeded to transcribe the change proposal. Thank you very much.
Hypotheses 1A, 1B, and 1C presented in the "Objective" section (lines 102-111) are identical. This needs to be corrected.
Hypothesis 1(A): The empowerment of the nurse plays a positive moderator role on the relationship between leader-member exchange and the organizational citizenship behaviour.
Hypothesis 1(B): The organizational support perceived by the nurse plays a positive moderator role on the relationship between leader-member exchange and the organizational citizenship behaviour.
Hypothesis 1(C): The leader-leader exchange plays a positive moderator role on the relationship between leader-member exchange and the organizational citizenship behaviour.